# Sustained Consumption of a Decaffeinated Green Coffee Nutraceutical Has Limited Effects on Phenolic Metabolism and Bioavailability in Overweight/Obese Subjects

**DOI:** 10.3390/nu14122445

**Published:** 2022-06-13

**Authors:** Miguel Ángel Seguido, Rosa Maria Tarradas, Susana González-Rámila, Joaquín García-Cordero, Beatriz Sarriá, Laura Bravo-Clemente, Raquel Mateos

**Affiliations:** Department of Metabolism and Nutrition, Institute of Food Science, Technology and Nutrition (ICTAN-CSIC), Spanish National Research Council (CSIC), José Antonio Nováis 10, 28040 Madrid, Spain; m.seguido@ictan.csic.es (M.Á.S.); rosa.tarradas.valero@gmail.com (R.M.T.); s.gonzalez@ictan.csic.es (S.G.-R.); j.garcia@ictan.csic.es (J.G.-C.); beasarria@ictan.csic.es (B.S.); lbravo@ictan.csic.es (L.B.-C.)

**Keywords:** green coffee, hydroxycinnamates, bioavailability, pharmacokinetics, microbial catabolites, biotransformation pathways

## Abstract

Knowledge on the bioavailability of coffee (poly)phenols mostly come from single dose postprandial studies. This study aimed at investigating the effects of regularly consuming a green coffee phenolic extract (GCPE) on the bioavailability and metabolism of (poly)phenols. Volunteers with overweight/obesity consumed a decaffeinated GCPE nutraceutical containing 300 mg hydroxycinnamates twice daily for two months. Plasma and urinary pharmacokinetics, and fecal excretion of phenolic metabolites were characterized by LC-MS-QToF at weeks 0 and 8. Fifty-four metabolites were identified in biological fluids. Regular consumption of the nutraceutical produced certain changes: reduced forms of caffeic, ferulic and coumaric acids in urine or 3-(3′-hydroxypenyl)propanoic, and 3,4-dihydroxybenzoic acids in feces significantly increased (*p* < 0.05) after 8 weeks; in contrast, coumaroylquinic and dihydrocoumaroylquinic acids in urine decreased (*p* < 0.05) compared to baseline excretion. The sum of intestinal and colonic metabolites increased after sustained consumption of GCPE, without reaching statistical significance, suggesting a small overall effect on (poly)phenols’ bioavailability.

## 1. Introduction

Green coffee beans are rich in phenolic compounds, mainly hydroxycinnamate esters, collectively known as hydroxycinnamic or chlorogenic acids [1], as opposed to roasted coffee where most phenolic compounds are lost during roasting or transformed and incorporated into melanoidins [2]. Phenol-rich extracts obtained from unroasted coffee beans (green coffee phenolic extracts, GCPE) are commonly used as dietary supplements or nutraceuticals. Consumption of GCPE is increasing due to the reported health benefits of green coffee (poly)phenols, with special emphasis on their effects on diabetes and obesity in humans [3], preventing insulin resistance, reducing appetite [4], regulating body weight [5], and favoring weight loss [6]. Green coffee phenolic compounds have also shown cardioprotective properties due to their antihypertensive, antihyperlipidemic, antifibrotic, or anti-inflammatory activities [7,8], even showing antiproliferative and cytotoxic effects in human cancer cell lines [9].

The metabolism, pharmacokinetics, and bioavailability of coffee phenols have been reported in several studies [10,11,12], in particular, green coffee hydroxycinnamic acids have proven to be extensively metabolized and partially absorbed in healthy subjects after acute consumption of a green/roasted coffee brew, with long permanence in the host, thus favoring their bioactivity [13]. In fact, some of these phenols can be directly absorbed in the small intestine, although most reach the large intestine where they are catabolized by the intestinal microbiota. These catabolites can exert specific functions, although the physiological role of phenols may be altered by the biotransformation of parent compounds [14]. It is important to bear in mind that the interaction between (poly)phenols and microbiota is bidirectional. The microbiota may act on (poly)phenols to enhance their conversion into active and bioavailable microbial catabolites or “postbiotics” [15,16,17]. Reciprocally, (poly)phenols may also modulate the gut microbiota ecology, exerting a prebiotic-like action [15,16], which might potentially affect the microbial catabolism of phenolic compounds.

Up to date, most studies on the bioavailability and metabolism of phenolic compounds are single dose, acute, postprandial assays with plasma and urine sample collection up to 12–24 h after intake of the test food or supplement. However, the long-term effects of regular consumption on the bioavailability of (poly)phenols have been scarcely studied so far. Only in Mena et al., (2021) [18] has the absorption, metabolism, and pharmacokinetic profile of roasted coffee phenolics after one month of repeated consumption of one or three cups of coffee or a confectionary containing coffee and cocoa been assessed. However, in this work, the authors did not study the bioavailability of coffee phenolics at the beginning of the intervention, and thus the final results could not be compared with the initial metabolism to assess potential modifications over prolonged consumption. In a previous study, these authors did study the effect of the sustained consumption of tablets containing green tea and green coffee phenolic extracts for 8 weeks, finding important differences in the urinary excretion of flavanol-derived catabolites, but less remarkable changes in the excretion of metabolites derived from green coffee hydroxycinnamates [19].

Bearing in mind the above points, the initial hypothesis of the present study was that the sustained intake of green coffee (poly)phenols may modify the overall bioavailability and metabolism of these compounds, likely due to the prolonged exposure of the intestinal microbiota, which would lead to a higher and more uniform concentration of the associated catabolites in the gastrointestinal tract and in plasma after absorption. In addition, considering the increasing consumption of GCPE as dietary supplements or nutraceuticals, usually aimed at weight management, it is important to understand the bioavailability and metabolism of phenols in the GCPE nutraceutical in the target population (i.e., subjects with obesity/overweight), which might differ from that in a beverage brewed from the green coffee beans. Therefore, the present work aimed at studying the possible changes in the absorption and metabolism profiles of hydroxycinnamates in plasma, urine, and feces after regular consumption of a GCPE nutraceutical by subjects with overweight/obesity, comparing the absorption, metabolism, and pharmacokinetic profile at baseline and after eight weeks of daily intake.

## 2. Materials and Methods

### 2.1. Chemicals

Ascorbic, 5-caffeoylquinic acid, 3′,4′-dihydroxycinnamic acid, 4′-hydroxy-3′-methoxycinnamic acid, 4′-hydroxycinnamic acid; hippuric acid, 3-(3′,4′-dihydroxyphenyl)propanoic acid, 3-(4′-hydroxy-3′-methoxyphenyl)propanoic acid, 3-(4′-hydroxyphenyl)propanoic acid, 3-(3′-hydroxyphenyl)propanoic acid, 3′,4′-dihydroxyphenylacetic acid, 4′-hydroxy-3′-methoxyphenylacetic acid, 4′-hydroxyphenylacetic acid, 3′-hydroxyphenylacetic acid, 3,4-dihydroxybenzoic acid, 4-hydroxy-3-methoxybenzoic acid, 4-hydroxybenzoic acid, and 3-hydroxybenzoic acid were purchased from Sigma-Aldrich (Madrid, Spain). 3,5-dicaffeoylquinic acid was purchased from PhytoLab (Vestenbergsgreuth, Germany). HPLC-grade solvents were acquired from Panreac (Madrid, Spain). Ultrapure water from MilliQ system (Millipore, Bedford, MA, USA) was used throughout the experiment.

### 2.2. Phenolic Composition of the GCPE

The green coffee phenolic extract (GCPE) was provided by Quimifarma Laboratorios S.L. (Toledo, Spain), which was a decaffeinated, phenol-rich (45%) commercial extract. The phenolic content and composition of the GCPE was verified by in-house HPLC-DAD analysis, as in previous studies [1,13]. Briefly, one gram of GCPE was dissolved in 100 mL of MilliQ water and filtered through a PVDF 0.45 µm filter. Phenols were separated on a Superspher 100 RP18 column (4.6 mm × 250 mm i.d., 4 μm; Agilent Technologies, Santa Clara, CA, USA) preceded by an ODS RP18 guard column kept in a thermostatic oven at 30 °C using an Agilent 1200 series liquid chromatographic system equipped with an autosampler, quaternary pump, and diode-array detector (DAD). An aliquot (20 µL) was eluted at a flow rate of 1 mL/min using a mobile phase of 1% formic acid in deionized water (solvent A) and acetonitrile (solvent B). The solvent gradient changed from 10% to 20% solvent B over 5 min, 20% to 25% solvent B over 30 min, 25% to 35% solvent B over 10 min, and then was maintained isocratically for 5 min, returning to the initial conditions over 10 min. Chromatograms were recorded at 320 nm, and 5-caffeoylquinic and 3,5-dicaffeoylquinic acids were used to calculate the mono- and di-acylcinnamate esters content, respectively, by external standard method. Samples were analyzed in triplicate.

The phenolic composition of the GCPE is detailed in Appendix A. Caffeoylquinic and dicaffeoylquinic acids were the main phenolic compounds detected in the extract, accounting for 85% of the quantified hydroxycinnamates, followed by 7.6% feruloylquinic and 4.6% of caffeoylferuloylquinic acids. The nutraceutical was prepared considering the total phenolic content of the GCPE (458 mg/g) so that each sachet contained 0.66 g of GCPE, providing 300 mg of phenolic compounds. Excipient (whey protein) and aroma compounds (two flavors were available, vanilla and chocolate) were added to the formulation to facilitate dispersion in water and increase acceptability.

### 2.3. Participants, Study Design, and Sample Collection

This study is part of a larger intervention carried out in 29 subjects with overweight or obesity designed to assess the effect of sustained consumption of GCPE on weight control, blood pressure, lipid metabolism, and glucose homeostasis. Of these participants, nine volunteers (8 men and 1 woman), aged 44 ± 9 years and with a body mass index (BMI) of 31 ± 4 agreed to be included in the present bioavailability study. Details of the recruitment process, inclusion and exclusion criteria, and ethical clearance are reported elsewhere [20]. Subjects signed an additional specific written informed consent to participate in the present bioavailability study. The main biochemical and vital characteristics of the volunteers at baseline (week 0) and at the end of the intervention (week 8) are shown in Appendix A. No adverse events were reported.

Participants attended the Human Nutrition Unit (HNU) at the Institute of Food Science, Technology and Nutrition (ICTAN-CSIC) on two consecutive days at the beginning of the intervention (week 0) and also on two consecutive days at the end of the study, after consuming daily two doses of the GCPE nutraceutical, providing 600 mg/d phenols, during two months (week 8). On each occasion, on the first visit day (Figure 1), subjects arrived at the HNU after an overnight (12 h) fast and provided a urine sample. A licensed health care professional placed an intravenous catheter in the subjects′ nondominant arm and collected a fasting blood sample (baseline, 0 h). Then, subjects drank the soluble GCPE nutraceutical dissolved in 200 mL of water. Subsequently, blood samples were collected in EDTA-coated tubes at 0.5, 1, 1.5, 2, 3, 4, 6, 8, 10, and 11 h. On the sampling days, volunteers remained at the HNU, where a (poly)phenol-free diet was provided. Half an hour after consuming the nutraceutical, volunteers ate a breakfast consisting of ham, cheese, white bread, and a yogurt. Lunch (paella, bread, and yogurt) and an afternoon snack (muffin and yogurt) were eaten 5 and 9 h, respectively, after consuming the nutraceutical. Water and isotonic beverages were freely available. Volunteers were instructed to eat a (poly)phenol-free dinner that night and to attend the HNU on the next day after a 12 h fast. Then, a final blood sample was obtained from the antecubital vein 24 h after consuming the nutraceutical. The same protocol was followed during visits at the end of the 8-week intervention. On each occasion, two days before each visit to the HNU, subjects consumed a low (poly)phenol diet free of foods containing hydroxycinnamates such as coffee, herbal teas, red wine, artichokes, whole grain cereals, vegetables, fruits and fruit juices, and dried fruits. Only banana, watermelon, cantaloupes, and potatoes were allowed.

Blood samples were immediately centrifuged (10 min, 3000 rpm, 4 °C) to obtain plasma. Urine was collected in plastic flasks containing preservative (0.5 g ascorbic acid) and sampled at different intervals: 0–3 h, 3–6 h, 6–9 h, 9–11 h, and 11–24 h after GCPE consumption, measuring total urine volume at each interval. Fecal samples were provided at baseline (−2 to 0 h) and 24 h after ingestion of the nutraceutical at the beginning and end of the intervention. All biological samples were stored at −80 °C until analysis. 

### 2.4. Plasma, Urine and Fecal Samples Processing and Analysis by HPLC-ESI-QToF

To extract plasma metabolites, 10 µL of 50% (*v*/*v*) aqueous formic acid, and 900 µL cold acetonitrile containing 50 µL of 10% (*w*/*v*) ascorbic acid were added to 400 µL defrosted plasma. Samples were vortexed and centrifuged (14,000 rpm, 10 min, 4 °C), repeating the extraction and combining the two supernatants, which were reduced to dryness under nitrogen [13]. Dried samples were resuspended in 150 µL of 0.1% aqueous formic acid (containing 10% acetonitrile acidified with 0.1% formic acid), centrifuged (15 min, 14,000 rpm, 4 °C), and the supernatant collected. Urine samples were diluted 1:1 with Milli-Q water and centrifuged (14,000 rpm, 20 min, 4 °C) prior to analysis. Phosphate-buffered saline (3 mL) was added to a representative 300 mg sample of thawed feces, homogenized in an ultrasound bath during 10 min, centrifuged (14,000 rpm, 20 min, 4 °C), and the supernatants collected. 

An amount of 30 µL of plasma and 5 µL of urine and fecal extracts, all previously filtered through 0.45 μm cellulose acetate membrane filters, were injected into an Agilent 1200 series LC system coupled to an Agilent 6530A Accurate-Mass Quadrupole Time-Of-Flight (Q-ToF) with ESI-Jet Stream Technology (Agilent Technologies, Santa Clara, CA, USA). A reverse-phase Ascentis Express C18 (15 cm × 3 mm, 2.7 µm) column (Sigma-Aldrich Quimica, Madrid, Spain), preceded by a Supelco 55215-U guard column (3 mm × 5 mm, 2.7 µm), was used for separation. Mobile phase A was 0.1% formic acid in Milli-Q water, and mobile phase B was acetonitrile containing 0.1% formic acid at a 0.3 mL/min flow rate. Details of the solvent gradient and Q-ToF acquisition conditions are given in the Appendix A. Metabolites were identified based on their retention time using authentic standards when possible. Those metabolites for which there were no available standards were tentatively quantified using the calibration curves of their corresponding phenolic precursors, as specified in the Appendix A, which contains additional information on the preparation and validation of calibration curves. Urine concentration of excreted metabolites was normalized to the total volume excreted in each studied interval. 

### 2.5. Pharmacokinetic and Statistical Analysis

The PKsolver add-on program was used to perform pharmacokinetic analyses in Microsoft Excel, including maximum concentration (C_max_), area under curve between 0–24 h (AUC_0–24_), and time to reach maximum concentration (T_max_) of metabolites in plasma and urine. All the results were statistically analyzed with SPSS software (version 27.0; SPSS, Inc., IBM Company, Armonk, NY, USA). The Shapiro–Wilk test was used to assess data normality. In view of the lack of normality, and considering the small sample size, comparisons between week 0 and week 8 were performed by the nonparametric Wilcoxon test for paired comparisons. The level of significance was *p* < 0.05. All data are expressed as mean ± standard error of the mean (SEM) unless specified otherwise.

## 3. Results

### 3.1. Identification and Quantification of Plasma Metabolites

Twenty-eight compounds derived from GCPE hydroxycinnamate intake were identified and quantified in the 0–24 h plasma samples. The identification of native compounds in the administered nutraceutical facilitated the targeting of their potential metabolites by searching for their exact mass and confirmation by fragmentation patterns (MS/MS). Appendix A shows the retention time (RT), molecular formula, accurate mass of the quasimolecular ion [M-H]^−^ after negative ionization, and the MS^2^ fragments of the main compounds identified in plasma using LC-QToF. Peak plasma concentrations (C_max_), time to reach C_max_ (T_max_), and area under the curve (AUC_0–24h_) values of the 28 metabolites detected in plasma at baseline and after consuming the GCPE product for 8 weeks are detailed in Table 1. Figure 2 presents the plasmatic pharmacokinetic profiles for some representative metabolites. 

Some unmetabolized compounds originally contained in the nutraceutical (5-caffeoylquinic acid (Figure 2A), 4- and 5-feruloylquinic acids) were also detected in plasma, as well as hydroxycinnamic acids (3′,4′-dimethoxycinnamic acid, 4′-hydroxy-3′-methoxycinnamic acid (*Ferulic acid*, *FA*), and 3′-hydroxy-4′-methoxycinnamic acid (*isoFerulic acid*, *iFA*) (Figure 2B)) resulting from the hydrolysis of their respective monoacylquinic esters and/or methylation of 3′,4′-dihydroxycinnamic acid (*Caffeic acid*). Some of these compounds were extensively metabolized into phase II sulfated and glucuronidated derivatives (3′-methoxycinnamic acid-4′-glucuronide (*FA-4′-glucuronide*) and 3′-methoxycinnamic acid-4′-sulfate (*FA-4′-sulfate*) (Figure 2C)). No free 4′-hydroxycinnamic acid (*Coumaric acid*, *CoA*) was detected, but its conjugated forms were present in plasma. According to the time of appearance in plasma, this group of metabolites showed early absorption in the small intestine, with T_max_ values between 0.5 and 2.1 h after GCPE intake (Table 1). Nevertheless, despite having a similar pharmacokinetic profile to these earlier-absorbed metabolites, FA-4′-sulfate and FA-4′-glucuronide showed a second C_max_ peak in plasma around 5–8 h after ingestion, displaying a biphasic plasma profile (Figure 2C), as well as sulfated and glucuronidated forms of CoA. Overall, these compounds had low concentrations in plasma, showing C_max_ values from 8 to 39 nM (Table 1). Comparing C_max_, T_max_ and AUC_0–24h_ values at the beginning of the trial (week 0) with those obtained after 8-week supplementation with the GCPE nutraceutical, FA values were significantly different between the two time points. Thus, a shortening of T_max_ from 1.8 h to 0.6 h was observed at week 0 vs. week 8, while C_max_ values increased from traces to 8 nM after supplementation, as well as its AUC_0–24h_ values from 34 nM min^−1^ at week 0 to 90 nM min^−1^ at the end of the intervention (Table 1). Similarly, AUC_0–24h_ also increased for *i*FA from 25 to 43 nM min^−1^. Moreover, T_max_ values for 5-caffeoylquinic showed a significant reduction after 8 weeks of GCPE supplementation (from 1.3 to 0.7 h).

Another important group of metabolites found in plasma corresponded to the reduced forms of hydroxycinnamic acids, such as 3-(3′,4′-dihydroxyphenyl)propanoic acid (*Dihydrocaffeic acid*, *DHCA*) (Figure 2D), 3-(4′-hydroxy-3′-methoxyphenyl)propanoic acid (*Dihydroferulic acid*, *DHFA*) (Figure 2E), 3-(3′-hydroxy-4′-methoxyphenyl)propanoic acid (*Dihydroisoferulic acid*, *DHiFA*), and 3-(3′,4′-dimethoxyphenyl)propanoic acid (*Dihydrodimethoxycinnamic acid*). All these metabolites were extensively transformed into their phase II derivatives: 3-(3′-hydroxyphenyl)propanoic acid-4′-sulfate (*DHCA-4′-sulfate*) (Figure 2F), 3-(3′-methoxyphenyl)propanoic acid-4′-glucuronide (*DHFA-4′-glucuronide*), 3-(3′-methoxyphenyl)propanoic acid-4′-sulfate (*DHFA-4′-sulfate*), 3-(4′-methoxyphenyl)propanoic acid-3′-glucuronide (*DHiFA-3′-glucuronide*), and 3-(4′-methoxyphenyl)propanoic acid-3′-sulfate (*DHiFA-3′-sulfate*). Feruloylglicine was also present in plasma at trace levels. These compounds appeared in plasma for longer than their precursors after GCPE consumption, reaching their C_max_ between 4 and 11 h postintake (Table 1), thus with kinetics compatible with colonic absorption. 

In addition, this group of metabolites showed higher C_max_ values than their precursors, being DHCA-4′-sulfate, DHFA, and DHCA the predominant metabolites (C_max_ values ranged from 220 nM to 340 nM, Table 1). No significant differences were observed for C_max_, T_max_, or AUC_0–24h_ values of each metabolite between baseline (week 0) and the end of the intervention (week 8).

Finally, several microbial metabolites identified as hydroxyphenylacetic, hydroxybenzoic, and hydroxyhippuric acid derivatives were also detected in plasma, with 4′-hydroxyphenylacetic and 4-hydroxy-3-methoxybenzoic acids being the most abundant catabolites in this matrix. Some changes were observed between week 0 and 8 with slightly higher C_max_ values at the end of the intervention, but these were not statistically significant (Table 1). Similarly, with the exception of 3-hydroxybenzoic acid, which T_max_ decreased from 8 h to 4 h, no significant differences were observed in pharmacokinetic parameters after 8 weeks of nutritional intervention in this group of microbial catabolites. 

### 3.2. Identification and Quantification of Urinary Metabolites

A total of 46 metabolites derived from GCPE consumption were identified in urine, 22 of which were also detected in plasma. Compound identification features are shown in Appendix A. Pharmacokinetic parameters (C_max_, T_max_ and AUC_0–24h_) at the beginning and at the end of the 8-week supplementation period are shown in Appendix A. The amounts excreted at the different collection intervals, before and after 8 weeks of regular nutraceutical consumption, are given in Appendix A, respectively. The total urinary recovery from 0 to 24 h is shown in Table 2 and the excretion percentages, together with some significant changes in representative metabolites, are summarized in Figure 3.

Unmetabolized hydroxycinnamoylquinic acids (3-, 4-, 5-caffeoylquinic, 3-, 4-, 5-feruloylquinic and coumaroylquinic acids) represented a minor part (1.3%) of the total polyphenols excreted in urine at baseline, decreasing to 1.1% at week 8 (Figure 3). In addition, this group of compounds showed the earliest absorption, since their C_max_ appeared between 0–6 h postintake (Appendix A). Nevertheless, the only statistically significant difference between week 0 and week 8 in this group of metabolites corresponded to the total urinary recovery (collected from 0 to 24 h after product intake) of coumaroylquinic acid, which decreased from 0.14 μmoles at week 0 to 0.10 μmoles after nutraceutical intervention (week 8) (Table 2). 

Lower amounts (0.9%) of free hydroxycinnamic acids, such as 3′,4′-dihydroxycinnamic acid (*Caffeic acid*, *CA*) or *iFA* were also detected at baseline (week 0) and at week 8 (Figure 3). In turn, their phase II derivatives (4′-hydroxycinnamic acid-3′-sulfate (*CA-3′-sulfate*), FA-4′-glucuronide, FA-4′-sulfate, 4′-methoxycinnamic acid-3′-glucuronide (*iFA-3′-glucuronide*) and 4′-methoxycinnamic acid-3′-sulfate (*iFA-3′-sulfate*)) accounted for 19.4% and 18.1% of phenolic excretion at week 0 and 8, respectively (Figure 3A). Actually, FA-4′-sulfate was the most abundant compound excreted both at baseline (32 µmol/24 h) and after the intervention (33 µmol/24 h) (Table 2). All these metabolites (hydroxycinnamates, hydroxycinnamic acids, and their phase II derivatives) were mainly excreted in the interval from 0 to 3 h, pointing to their early absorption in the small intestine. However, glucuronidated and sulfated FA derivatives were also excreted in great amounts between 11 and 24 h, emphasizing the biphasic profile observed in plasma (Figure 2C). C_max_ and T_max_ values showed no significant changes in these compounds between week 0 and 8. As for the AUC_0–24h_, there was a statistically significant increase of the AUC_0–24h_ values of *i*FA-3-sulfate and *iFA-*3-glucuronide after sustained consumption of the nutraceutical (Appendix A). Total amounts excreted of these early-absorption metabolites were 50 µmol (21.6% of the phenolic excretion) and 59 µmol (20.1%) before and after sustained consumption of GCPE, respectively (Table 2).

Reduced forms of hydroxycinnamic acids (DHCA, DHFA, DH*i*FA, 3-(4′-hydroxyphenyl)propanoic acid (*Dihydrocoumaric acid*, *DHCoA*), and dihydrodimethoxycinnamic acid), together with their sulfated and glucuronidated phase II derivatives, constituted the main group of phenolics excreted in urine (Figure 3A). Feruloylglicine and *iso*feruloylglicine were also detected, showing delayed kinetics compatible with colonic absorption (Appendix A), with feruloylglycine being one of the most abundant metabolites in this group (with excretions of 19 and 25 µmol in 24 h at baseline and after 8 weeks of supplementation, respectively). Lastly, reduced forms of hydroxycinnamoylquinic acids (3-, 4- and 5-dihydrocaffeoylquinic acids, 3-, 4-, and 5-dihydroferuloylquinic acids and two isomers of dihydrocoumaroylquinic acid) were also identified in urine, although with lower abundance (0.8% of total phenolics at week 0 and 0.6% at week 8, Figure 3A). Taken together, all these metabolites showed delayed kinetics compatible with colonic absorption (high excretion between 6 and 24 h postintake, Appendix A) and amounted up to 45.7% and 48.4% (week 0 and week 8, respectively) of the total compounds quantified in urine. DHCA stands out among the most abundant catabolites in urine, which excretion significantly increased from 16 to 31 µmol/24 h after the nutritional intervention (Table 2, Figure 3B). The concentration of other metabolites also increased after 8 weeks, such as DHCA-4′-sulfate (which increased from 8 to 10 µmol/24 h), DHFA-4′-glucuronide (from 5 to 7 µmol/24 h), and DHCoA (from 4 to 6 µmol/24 h) (Figure 3B). In turn, urinary concentration of dihydrocaffeoylquinic acid and dihydrocoumaroylquinic acid significantly decreased from 0.35 to 0.27 and from 0.22 to 0.17 µmol/24 h after the intervention, respectively (Table 2). Therefore, quantitative changes between baseline and week 8 did not follow a clear pattern, probably due to the high interindividual variability observed. However, there was a trend towards an increase in urinary elimination of colonic-absorption metabolites (from 106 μmoles/24 h at week 0 to 132 at week 8), as well as an apparent prevalence of sulfated forms over glucuronidated metabolites (Table 2). It is worth pointing out that some metabolites, such as DHCA, DHCA-4′-sulfate, DHCoA, or DHFA-4′-glucuronide, were present in the basal urine samples obtained before the nutraceutical intake (Appendix A), with excretion at 0 h significantly higher at the end of the intervention (week 8) compared to baseline (week 0, *p* < 0.01). In terms of pharmacokinetic parameters (Appendix A), all the colonic compounds showed T_max_ values mainly between 9 and 24 h. However, at the end of the intervention, T_max_ values tended to decrease, with these changes being statistically significant (*p* < 0.05) for sulfate and glucuronide conjugates of DHCoA. In addition, when the AUC_0–24h_ values from week 0 and week 8 were compared, statistically significant changes (*p* < 0.05) were also found for DHCA (increasing from 96 to 125 μM min^−1^), 3-(phenyl)propanoic acid-4′-sulfate (*DHCoA-4′-sulfate*) (from 161 to 251 μM min^−1^), and 3-dihydroferuloylquinic acid (from 1.6 to 4 μM min^−1^). Regarding C_max_, only DHCA values significantly increased at the end of the intervention (from 7.7 to 10 μM).

Finally, some microbial metabolites, such as derivatives of hydroxyphenylacetic and hydroxbenzoic acids, along with hydroxyhippuric acid, could also be detected in urine in high concentrations (Table 2). These metabolites accounted up to 32.7% and 31% of the total phenolics at week 0 and 8, respectively (Figure 3A), showing extensive excretion in the last collection period (11–24 h) in accordance with their microbial origin. Among them, only 3′,4′-dihydroxyphenylacetic acid and 3′-hydroxyhippuric acid showed a statistically significant increase after the intervention (Table 2, Figure 3B). In turn, 4-hydroxybenzoic acid decreased from 1.3 to 1.01 µmol/24 h at week 0 and 8, respectively. No significant differences were observed in the pharmacokinetic parameters or in the total excretion of these microbial metabolites.

In summary, the total amount of hydroxycinnamate metabolites excreted in 24 h urine reached 231 µmol at week 0 and 274 µmol after 8 weeks of daily intake of the nutraceutical (Table 2), which represents, respectively, 27.3% and 32.3% of the 847 µmol (300 mg) of phenols consumed. Although the total amount of urinary metabolites was higher at week 8, it was not statistically different compared to baseline (week 0).

### 3.3. Identification and Quantification of Fecal Metabolites

As shown in Table 3, eighteen hydroxycinnamic acid derivatives were quantified in feces collected at 0 and 24 h after consuming the GCPE nutraceutical at week 0 and after 8 weeks of daily supplementation. These compounds included minor amounts of hydroxycinnamates and hydroxycinnamic acids, like 3-, 4-, and 5-feruloylquinic acids, CA, FA, and CoA, in which the total concentrations after 8 weeks of intervention were higher both at 0 and 24 h, but without reaching statistical significance. 

Regarding colonic metabolites, some compounds already quantified in plasma and urine were also present in feces, such as, DHFA, DHCoA, and 5-dihydrocaffeoylquinic acids. In addition, unconjugated 3-(3′-hydroxyphenyl)propanoic acid (*Dihydroisocoumaric acid, DHiCoA*) was also found in fecal samples, which corresponded to one of the main metabolites excreted in feces. Of note, the concentration of this metabolite in samples collected at 0 h increased from 0.21 µmol/g at week 0 to 0.4 µmol/g at week 8, without reaching statistical significance (*p* > 0.05). This group of colonic metabolites accounted for 39.7% of the total metabolites excreted in the samples collected at week 0 and increased up to 66.7% after 8 weeks of intervention, respectively, which might suggest an accumulative effect derived from the sustained daily intake of the nutraceutical. 

Finally, another large group of phenolic metabolites present in feces were the microbial metabolites, including 3′,4′-dihydroxyphenylacetic acid, 4′-hydroxy-3′-methoxyphenylacetic acid, 3′-hydroxyphenylacetic acid, 3,4-dihydroxybenzoic acid, 4-hydroxy-3-methoxybenzoic acid, and 3- and 4- hydroxybenzoic acids (Table 3). This group constituted 59.1% and 51.5% of the total metabolites found at week 0 in the samples collected before and 24 h after consuming the nutraceutical, respectively. Contrary to what was observed for the colonic metabolites, the excretion of these catabolites was lower at the end of the intervention, amounting to only 30.5% and 32.9% of the total phenolics in 0 and 24 h feces, respectively, at week 8. In this group, 3′-hydroxyphenylacetic acid and 3,4-dihydroxybenzoic acids were the most abundant compounds, and particularly the latter showed a significant increase at 0 h, from 0.016 µmol/g to 0.026 µmol/g at week 0 and after the 8-week intervention, respectively (*p* < 0.05). In turn, excretion of 3′-hydroxyphenylacetic acid decreased from 0.31 µmol/g at week 0 to 0.18 µmol/g at week 8 (samples collected at 0 h; concentration in samples collected at 24 h decreased from 0.21 to 0.11 µmol/g, at week 0 and week 8, respectively).

## 4. Discussion

Nutraceuticals based on (poly)phenols-rich vegetable extracts, such as green coffee, are widely used to combat overweight, obesity, and associated diseases [6]. In this context, understanding the bioavailability and metabolic fate of dietary phenols are key to better know about their potential effects on human health [21]. However, it is still poorly understood how sustained consumption of phenolic compounds may affect their absorption and metabolism, as most bioavailability studies have been acute assays that determined plasma and/or urinary metabolites up to 12–24 h after consuming a single dose of the nutraceutical or food containing the bioactive compound(s). Therefore, the present work aimed to assess the bioavailability of green coffee hydroxycinnamates at the beginning and after 8 weeks of sustained consumption of a GCPE nutraceutical in order to explore any potential adaptive effect in the metabolic profile of this type of phenolic compounds after long-term consumption. 

Quantitatively, results are in agreement with those previously obtained in our research group with a green/roasted coffee blend [13], showing that hydroxycinnamate esters were partially absorbed and extensively metabolized in the intestinal tract, with the gut microbiota playing an important role in catabolism. Total urinary excretion of absorbed metabolites accounted for 27.3% of the 847 µmol ingested at week 0, increasing up to 32.3% at the end of the 8-week intervention, with percentages in line with values reported by other authors after coffee consumption [13,22,23]. Qualitatively, no different metabolites were identified before and after regular consumption of the nutraceutical, suggesting no major changes on the biotransformation of phenolic compounds. This small increase in total urinary excretion of metabolites (only 5% between week 0 and week 8) was not statistically significant. However, considering the low bioavailability of green coffee hydroxycinnamates this increment is of interest, and it could be attributed to the sustained consumption of the nutraceutical. The question remains whether prolonging the intake of phenolic-rich foods might have a more relevant effect on the overall bioavailability or, on the contrary, saturation might occur, as suggested by Mena et al. in their repeated-dose study, where they observed a reduced bioavailability of coffee phenolic acids as the daily amount consumed during 4 weeks increased [18].

Unmetabolized parent compounds (caffeoyl-, feruloyl-, and *p*-coumaroylquinic acids), which were the major components of the nutraceutical (Appendix A), were detected in minor amounts in plasma (Table 1), urine (Table 2), and feces (Table 3), pointing to an extensive metabolization. Lower C_max_ values of 5-caffeoylquinic and 4-,5-feruloylquinic acids (from 22 to 34 nM) were observed in plasma (Table 1), peaking at short times (T_max_ between 0.9 and 1.3 h); they were also excreted in the urine, along with other monoacylquinic acids like coumaroylquinic acid, mainly within 0 to 3 h after consumption of the nutraceutical (Appendix A). These kinetics are compatible with that observed in previous studies [13,22], indicating a rapid absorption of these compounds in the small intestine. At this stage, the hydrolysis of monoacylquinic and diacylquinic acids were hydrolyzed by mammalian esterases, giving rise to free hydroxycinnamic acids (CA, FA and CoA), which are transformed into sulfated, glucuronidated, and methylated phase II derivatives by catechol-*O*-methyltransferase (COMT), sulfotransferases (SULT), or UDP-glucuronosyltransferases (UGT). All the aforementioned metabolites (hydroxycinnamates, hydroxycinnamic acids, and their phase II derivatives) reached the bloodstream and were excreted mainly at short times, between 0 and 3 h after intake, amounting up to 21.6% and 20.1% of all urinary metabolites excreted in 0–24 h at week 0 and at the end of the 8-week intervention, respectively. It is noteworthy that glucuronidated and sulfated forms showed a second excretion peak, from 6 up to 9 h, in agreement with the observed behavior in plasma (Figure 2C), which may be related to biphasic kinetics due to the enterohepatic circulation and/or colonic metabolism of the hydroxycinnamates [24]. Furthermore, the prevalence of feruloylquinic acid derivatives over caffeoylquinic forms is notorious; in fact, FA-4′-sulfate was one of the most abundant urinary metabolites, up to 32 µmol at week 0 and 33 µmol after the nutraceutical intervention. This suggests extensive methylation of caffeoylquinic acids via COMT [22], considering that these compounds were the most abundant in the GCPE nutraceutical (Appendix A). Although some significant changes were found in the pharmacokinetic parameters of FA, iFA, and 5-caffeoylquinic acid in plasma (Table 1), along with a reduction in coumaroylquinic acid in urine (Table 2), there were no qualitative differences or a clear pattern in this group of metabolites at the beginning vs. the end of the intervention, probably due to the large interindividual variability. 

Most of the hydroxycinnamic acid derivatives reached the colon and were substrates for microbial reductases before their absorption and conjugation into phase II metabolites. It is well known that microbiota esterases are able to hydrolyze the phenolic–quinic acid linkage, and then the released phenolic acids are converted into reduced forms (namely dihydroxycinnamic acids), which are absorbed through the colonic epithelium and transported via portal circulation to the liver [13,24]. These reduced forms can be conjugated by phase II enzymes, which results in a wide range of sulfated and glucuronidated microbial derivatives that reach the systemic circulation and, at the end, are excreted in the urine. It should be noted that among the colonic catabolites, sulfation was the predominant phase II transformation and, to a lower extent, glucuronidation, in agreement with Sanchez-Bridge et al. [25]. These colonic metabolites, together with other colonic ferulic acid derivatives such as feruloylglycine, and a minor group constituted by the dihydromonoacylquinic acids, formed the predominant group of metabolites in plasma and urine, accounting for 45.7% and 48.4% of total phenolics excreted at week 0 and after 8 weeks of intervention, respectively (Figure 3A). These results highlight once again the important role of the microbiota in the metabolism of hydroxycinnamates. Plasma concentration of reduced catabolites was higher than that of their parent compounds, with C_max_ values ranging from 60 to 340 nM, although no significant differences were found in any pharmacokinetic parameter when comparing week 0 vs. 8 (Table 1). One of the few statistically significant differences obtained in the present study was DHCA C_max_ in urine samples, which increased from week 0 to 8, along with DHCoA-4′-sulfate and 3-dihydroferuloylquinic acid, in which AUC_0–24h_ also increased significantly from week 0 to week 8 (Appendix A). In general, the pharmacokinetic parameters showed an upward trend after the 8-week green coffee nutraceutical intake. These results were in line with the higher amount of colonic metabolites quantified from 0–24 h at week 8 (136 μmoles) versus 106 µmol at week 0, although this difference was not statistically significant, in contrast to the individual amounts of DHCA, DHCoA, DHCA-4′-sulfate, and DHFA-4′-glucuronide, which increased significantly (Table 2, Figure 3B). It is worth noting that these metabolites, along with DHCA-3′-sulfate, DHCoA-4′-sulfate, and feruloylglycine, were the most abundant metabolites in urine. Interestingly, they were present at baseline (time 0 h before GCPE consumption), which could derive either from hydroxycinnamate consumption before the 48 h restriction, reinforcing the idea of their delayed elimination, or from the biotransformation of other phenolic compounds present in nonrestricted foods. Interestingly, DHCoA-4′-sulfate and feruloylglycine, along with unconjugated 3-(3′-hydroxyphenyl)propanoic acid (DH*i*CoA), were the only coffee-derived metabolites in which urinary excretion was increased in the study by Mena et al. after 8 weeks of daily consumption of tablets containing green tea and green coffee phenolic extracts [19]. In our study, only feruloylglycine excretion in urine was higher after 8 weeks of daily intake of the GCPE nutraceutical, yet without reaching the level of significance (*p* > 0.05) (Table 2).

Considering these results, future studies should extend the urine collection period beyond 24 h. Lastly, the higher basal excretion of this group of colonic metabolites at week 8 compared to week 0 (79 vs. 43 μmoles, respectively, Appendix A) might be affected by the previous dose of the nutraceutical taken at week 8, which was consumed only 17 h before sample collection. However, this group of compounds was present in low amounts in fecal samples obtained at week 0 and 8 (Table 3), probably due to extensive microbial catabolism to hydroxyphenylacetic acid and derived metabolites. Remarkably, 5-dihydrocaffeoylquinic acid, a high molecular weight compound and hallmark of hydroxycinnamate metabolism, was found in fecal samples too (Table 3).

On the other hand, high amounts of low molecular weight compounds, termed as ‘other microbial metabolites′ (hydroxyphenylacetic and hydroxybenzoic acid derivatives), have been quantified in plasma, urine, and feces, resulting from extensive colonic biotransformation. These compounds were present in the biological samples before nutraceutical consumption, since they are not exclusively derived from the catabolism of hydroxycinnamic acids and can participate in other biotransformation pathways. Among the significant increase of some of microbial metabolites in urine outstands 3′-hydroxyhippuric acid, which raised from 32 µmol at week 0 up to 46 µmol at week 8, becoming the most abundant metabolite quantified in urine (Table 2, Figure 3B). However, it cannot be considered a biomarker of coffee intake since it can derive from multiple biotransformation pathways, as just mentioned. Instead, the high levels of FA-4′-sulfate, DHCA, feruloylglycine, or DHCoA-4′-sulfate in urine could be positively correlated with the intake of chlorogenic acids. Therefore, these compounds might be proposed as biomarkers of coffee intake or compliance, and thus be used to assess adherence in intervention trials with coffee as well as a marker of habitual dietary intake of hydroxycinnamates in observational studies.

Overall, the concentration of phenolic metabolites excreted in urine increased from week 0 (231 µmol) to week 8 (274 µmol), without reaching the level of statistical significance (Table 2). This increase was slightly higher for colonic metabolites (which excretion was 24.5% higher in week 8 vs. week 0) compared to intestinal or other low molecular weight catabolites (that increased 18–20% in week 8 vs. week 0). However, this apparently higher absorption of colonic metabolites was not reflected in the circulating concentrations of these compounds in plasma (Table 1).

The amounts of plasmatic, urinary, and fecal metabolites after consumption of the GCPE nutraceutical varied noticeably between participants (detailed in Appendix A), showing no clear patterns in the excretion of phenolic metabolites among volunteers. On the contrary, each subject may show a different response depending on the biological sample analyzed, which increases the complexity of interindividual variability. This is in agreement with previous studies that highlight interindividual variability as an important factor affecting the bioavailability of (poly)phenols, pointing to differences in the intestinal microbiota of each person as a key factor, although understanding the underlying causes on interindividual variability is still incomplete and challenging [26]. Certainly, the microbial bioconversion capacity of each person is related to their specific microbiota composition, which influences the final metabolites produced, their bioavailability and biotransformation, and thus, the final impact on the health of the host [27,28]. For example, DHCA and DHFA, in addition to their precursors and other minor catabolites, are present in the upper regions of the large intestine, where they can act as antioxidants and prebiotics [29]. Nevertheless, the influence of other factors, such as age, sex, physiological status, diet, and dose, also need to be taken into account [19]. Therefore, more studies are needed to better understand the influence of these factors on the bioavailability of phenolic compounds.

This study has several limitations: it would have been interesting to extend urinary and fecal sample collection up to 48 h to recover delayed excreted metabolites; also, changes in the intestinal microbiota have not been addressed. In addition, the sample size was estimated following similar previous studies on the urinary excretion of phenolic compounds from coffee [13,19], but we do not perform power calculation due to the small number of studies available as well as the high interindividual variability observed in them. Strengths of the study: pharmacokinetic analysis of the long-term effects of coffee consumption on (poly)phenols bioavailability has been studied. Moreover, the collection of fecal samples enabled a more complete view of the bioavailability of hydroxycinnamic acids. To end, background features were controlled, such as volunteers′ physical activity and dietary habits. Lifestyle characteristics and socioeconomic status of the volunteers was homogenous. Intake of the nutraceutical at each visit was also monitored through the indicated markers.

In conclusion, this study confirms that phenolic compounds contained in the GCPE nutraceutical are highly metabolized throughout the gastrointestinal tract in a population of overweight and obese subjects, being differentially absorbed in the upper intestine compared to the colon. The colonic microbiota has played a key role in the metabolism of coffee hydroxycinnamates, since most of the metabolites characterized were formed at the colon. In addition, when the bioavailability before and after 8 weeks of daily consumption of the green coffee nutraceutical were compared, a higher trend in the absorption of GCPE was observed after regular consumption, but the metabolic profiles in plasma, urine, and feces did not statistically change. The present study contributes to better understand the effect of sustained consumption of a phenol-rich extract such as GCPE, since there are limited data on repeated exposure to phenols in bioavailability studies. This could help to develop refined dietary strategies and recommendations to optimize the beneficial effects of phenol-rich foods. Furthermore, FA-4′-sulfate, DHCA, feruloylglycine, or DHCoA-sulfate may be proposed as biomarkers of hydroxycinnamate intake, due to the high levels of these compounds in the biological fluids analyzed.

## Figures and Tables

**Figure 1 nutrients-14-02445-f001:**
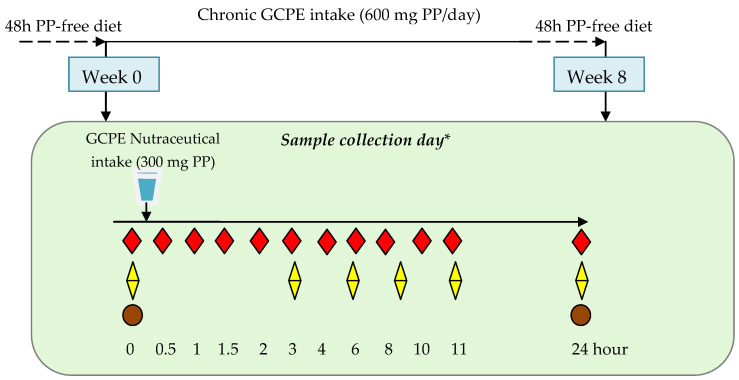
Schematic view of the study and collection of blood, urine, and fecal samples. ***** On each sampling day, volunteers consumed a single dose of the GCPE nutraceutical dissolved in water. GCPE: green coffee phenolic extract; PP: (poly)phenols.

**Figure 2 nutrients-14-02445-f002:**
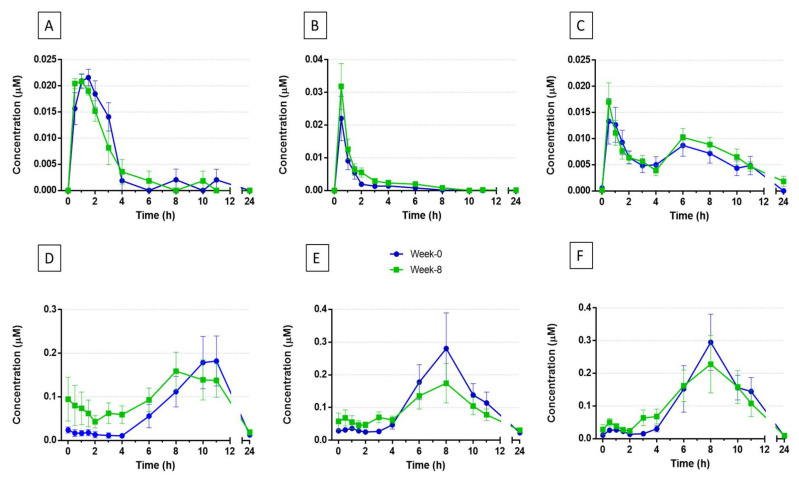
Plasma kinetic profile of (**A**) 5-caffeoylquinic acid, (**B**) isoferulic acid, (**C**) 3′-methoxycinnamic acid-4-sulfate, (**D**) 3-(3′,4′-dihydroxyphenyl)propanoic acid, (**E**) 3-(4′-hydroxy-3′-methoxyphenyl)propanoic acid, and (**F**) 3-(3′-hydroxyphenyl)propanoic acid-4′-sulfate after consumption of a nutraceutical containing 300 mg of hydroxycinnamic acids. Values are means ± SEMs (*n* = 9). Blue lines: week 0 of the intervention. Green lines: week 8 of the intervention.

**Figure 3 nutrients-14-02445-f003:**
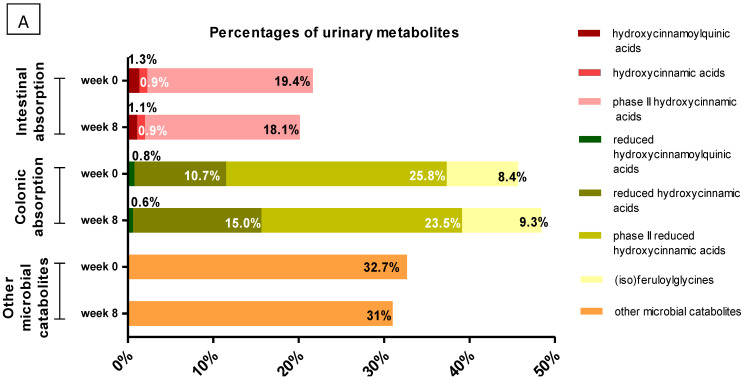
24 h cumulative urinary excretion represented as (**A**) percentages of the main groups of phenolic metabolites identified, and (**B**) selected metabolites with significant increases in their excretion at the beginning (week 0) and after 8 weeks consuming the GCPE nutraceutical. DHCA: Dihydrocaffeic acid; DHCoA: Dihydrocoumaric acid; DHCA-4′-sulfate: 3-(3′-Hydroxyphenyl)propanoic acid-4′-sulfate; DHFA-4′-glucuronide: 3-(3′-Methoxyphenyl)propanoic acid-4′-glucuronide. Values are means ± SEMs (*n* = 9). * *p* < 0.05 ** *p* < 0.01 week 0 vs. week 8.

**Table 1 nutrients-14-02445-t001:** Pharmacokinetics of plasma metabolites after consumption of the GCPE nutraceutical containing 300 mg of hydroxycinnamic acids at the beginning (week 0) and the end of the intervention (week 8).

	C_max_ (µM)		T_max_ (h) or Range ^a^		AUC_0–24h_ (µM min^−1^)	
Metabolite	Week 0	Week 8	Week 0	Week 8	Week 0	Week 8
**Intestinal absorption**						
5-Caffeoylquinic acid	0.023 ± 0.001	0.022 ± 0.001	1.3 ± 0.2 **	0.7 ± 0.1 **	0.08 ± 0.02	0.062 ± 0.009
5-Feruloylquinic acid	0.032 ± 0.004	0.034 ± 0.005	0.9 ± 0.1	0.9 ± 0.1	0.09 ± 0.02	0.08 ± 0.02
4-Feruloylquinic acid	0.029 ± 0.004	0.031 ± 0.004	0.9 ± 0.1	0.9 ± 0.1	0.06 ± 0.01	0.07 ± 0.01
4′-Hydroxy-3′-methoxycinnamic acid (*Ferulic acid, FA*)	Traces ^b,^**	0.008 ± 0.001 **	1.8 ± 0.6 *	0.6 ± 0.1 *	0.034 ± 0.006 **	0.09 ± 0.02 **
3′-Hydroxy-4′-methoxycinnamic acid *(isoFerulic acid*, *iFA*)	0.023 ± 0.006	0.032 ± 0.007	0.5 ± 0.1	0.5 ± 0	0.025 ± 0.007 **	0.043 ± 0.008 **
3′,4′-Dimethoxycinnamic acid	Traces ^b^	Traces ^b^	(2–4) ^a^	(2–4) ^a^	0.007 ± 0.003	0.01 ± 0.006
3′-Methoxycinnamic acid-4′-glucuronide (*FA-4′-glucuronide*)	0.016 ± 0.009	0.009 ± 0.002	1.2 ± 0.3 and 4.7 ± 0.3	1.2 ± 0.3 and 6.0 ± 0.7	0.06 ± 0.01	0.063 ± 0.02
3′-Methoxycinnamic acid-4′-sulfate (*FA-4′-sulfate*)	0.017 ± 0.004	0.017 ± 0.004	0.78 ± 0.09 and 6.8 ± 0.7	0.56 ± 0.06 and 7.2 ± 0.6	0.11 ± 0.03	0.13 ± 0.02
Cinnamic acid-4′-glucuronide (*CoA-4′-glucuronide)*	0.039 ± 0.002	0.038 ± 0.002	1.4 ± 0.3 and 7.0 ± 0.9	2.1 ± 0.2 and 9 ± 2	0.72 ± 0.08	0.71 ± 0.07
Cinnamic acid-4′-sulfate (*CoA-4′-sulfate)*	0.02 ± 0.02	0.03 ± 0.01	1.7 ± 0.9 and 6 ± 3	0 ± 0 and 11 ± 3	0.2 ± 0.2	0.4 ± 0.2
**Colonic absorption**						
3-(3′,4′-Dihydroxyphenyl)propanoic acid (*Dihydrocaffeic acid*, *DHCA*)	0.22 ± 0.06	0.22 ± 0.05	10.2 ± 0.3	9.4 ± 0.5	2.0 ± 0.6	2.1 ± 0.5
3-(4′-Hydroxy-3′-methoxyphenyl)propanoic acid (*Dihydroferulic acid*, *DHFA*)	0.3 ± 0.1	0.25 ± 0.06	7.6 ± 0.4	8.4 ± 0.6	2.2 ± 0.6	1.8 ± 0.3
3-(3′-Hydroxy-4′-methoxyphenyl) propanoic acid (*Dihydroisoferulic acid,* *DHiFA*)	0.07 ± 0.02	0.07 ± 0.02	7 ± 1	7 ± 1	0.5 ± 0.2	0.6 ± 0.2
3-(3′,4′-Dimethoxyphenyl)propanoic acid (*Dihydrodimethoxycinnamic acid*)	0.11 ± 0.02	0.12 ± 0.01	4 ± 1	10 ± 4	1.2 ± 0.1	1.3 ± 0.1
3-(3′-Hydroxyphenyl)propanoic acid-4′-sulfate (*DHCA-4′-sulfate*)	0.34 ± 0.09	0.33 ± 0.07	7.9 ± 0.5	8.4 ± 0.6	2.3 ± 0.5	2.1 ± 0.4
3-(3′-Methoxyphenyl)propanoic acid-4′-glucuronide (*DHFA-4′-glucuronide*)	0.13 ± 0.03	0.12 ± 0.02	8.7 ± 0.6	8.7 ± 0.6	1.1 ± 0.3	0.8 ± 0.1
3-(4′-Methoxyphenyl)propanoic acid-3′-glucuronide (*DHiFA-3′-glucuronide*)	0.06 ± 0.02	0.07 ± 0.03	9.0 ± 0.6	8.1 ± 0.8	0.6 ± 0.4	0.7 ± 0.4
3-(3′-Methoxyphenyl)propanoic acid-4′-sulfate(*DHFA-4′-sulfate*)	0.19 ± 0.08	0.17 ± 0.05	8.1 ± 0.4	8.4 ± 0.6	1.5 ± 0.7	1.1 ± 0.3
3-(4′-Methoxyphenyl)propanoic acid-3′-sulfate(*DHiFA-3′-sulfate*)	0.04 ± 0.01	0.11 ± 0.06	7 ± 1	11.6 ± 3.2	0.3 ± 0.1	0.8 ± 0.5
Feruloylglycine	Traces ^b^	Traces ^b^	7 ± 1	8.4 ± 0.6	0.029 ± 0.007	0.026 ± 0.004
**Other microbial metabolites**						
4′-Hydroxy-3′-methoxyphenylacetic acid	0.14 ± 0.06	0.12 ± 0.05	7 ± 4	4 ± 3	1.4 ± 0.9	1.5 ± 0.8
4′-Hydroxyphenylacetic acid	2 ± 1	7 ± 3	2 ± 2	0.6 ± 0.1	2 ± 1	36 ± 23
3′-Hydroxyphenylacetic acid	0.20 ± 0.07	0.15 ± 0.05	8 ± 2	5 ± 1	2.6 ± 0.9	1.6 ± 0.6
4-Hydroxy-3-methoxybenzoic acid	1.1 ± 0.2	1.6 ± 0.4	7 ± 3	7 ± 2	9 ± 2	14 ± 4
4-Hydroxybenzoic acid	0.09 ± 0.01	0.08 ± 0.01	7 ± 2	5 ± 3	0.9 ± 0.2	0.5 ± 0.2
3-Hydroxybenzoic acid	0.08 ± 0.02	0.10 ± 0.03	8 ± 1 *	4 ± 1 *	0.7 ± 0.3	0.8 ± 0.3
4′-Hydroxyhippuric acid	0.08 ± 0.01	0.09 ± 0.01	12 ± 3	8 ± 2	1.2 ± 0.2	1.3 ± 0.2
3′-Hydroxyhippuric acid	1.1 ± 0.3	1.3 ± 0.2	10.0 ± 0.4	9.2 ± 0.5	10 ± 3	13 ± 3

Values are means ± SEM (*n* = 9). * *p* < 0.05 ** *p* < 0.01 week 0 vs. week 8. C_max_, maximum plasma concentration; T_max_, time to reach the maximum plasma concentration; AUC_0–24h_, area under the curve. ^a^ Range where the metabolite showed the highest value. ^b^ At trace levels, pharmacokinetic parameters were not determined.

**Table 2 nutrients-14-02445-t002:** Cumulative excretion (0–24 h) of urinary metabolites after the intake of GCPE nutraceutical at the beginning (week 0) and the end of the intervention (week 8).

	Total 0–24 h (µmol)	
Metabolite	Week 0	Week 8
**Intestinal absorption**		
3-Caffeoylquinic acid	0.21 ± 0.05	0.18 ± 0.03
5-Caffeoylquinic acid	0.33 ± 0.06	0.33 ± 0.04
4-Caffeoylquinic acid	0.13 ± 0.07	0.15 ± 0.07
3-Feruloylquinic acid	0.29 ± 0.07	0.34 ± 0.07
5-Feruloylquinic acid	1.5 ± 0.2	1.5 ± 0.2
4-Feruloylquinic acid	0.4 ± 0.2	0.4 ± 0.1
Coumaroylquinic acid	0.14 ± 0.03	0.10 ± 0.02 *
3′,4′-Dihydroxycinnamic acid (*Caffeic acid*, *CA*)	0.42 ± 0.06	0.45 ± 0.09
4′Hydroxycinnamic acid-3′-sulfate (*CA-3′-sulfate*)	3.7 ± 0.5	4.4 ± 0.8
3′-Hydroxy-4′-methoxycinnamic acid (*isoFerulic acid*, *iFA*)	1.7 ± 0.2	2.1 ± 0.2
3′-Methoxycinnamic acid-4′-glucuronide (*FA-4′-glucuronide*)	3.4 ± 0.4	3.7 ± 0.5
4′-Methoxycinnamic acid-3′-glucuronide (*iFA-3′-glucuronide*)	4.4 ± 0.5	6 ± 1
3′-Methoxycinnamic acid-4′-sulfate (*FA-4′-sulfate*)	32 ± 5	33 ± 5
4′-Methoxycinnamic acid-3′-sulfate (*iFA-3′-sulfate*)	1.1 ± 0.3	2.1 ± 0.9
**TOTAL—Intestinal metabolites**	50 ± 6	59 ± 8
**Colonic absorption**		
3-(3′,4′-Dihydroxyphenyl)propanoic acid (*Dihydrocaffeic acid*, *DHCA*)	16 ± 2	31 ± 5 **
3-(4′-Hydroxy-3′-methoxyphenyl) propanoic acid (*Dihydroferulic*, *DHFA*)	0.5 ± 0.2	1.1 ± 0.4
3-(3′-Hydroxy-4′-methoxyphenyl) propanoic acid (*Dihydroisoferulic*, *DHiFA*)	3.7 ± 0.4	3.2 ± 0.2
3-(4′-Hydroxyphenyl)propanoic acid (*Dihydrocoumaric acid*, *DHCoA*)	4 ± 1	6 ± 2 **
3-(3′,4′-Dimethoxyphenyl)propanoic acid (*Dihydrodimethoxycinnamic acid*)	0.7 ± 0.2	0.63 ± 0.08
3-(4′-Hydroxyphenyl)propanoic acid-3′-glucuronide (*DHCA-3′-glucuronide*)	0.6 ± 0.2	0.5 ± 0.1
3-(3′-Hydroxyphenyl)propanoic acid-4′-sulfate (*DHCA-4′-sulfate*)	8 ± 2	10 ± 2*
3-(4′-Hydroxyphenyl)propanoic acid-3′-sulfate (*DHCA-3′-sulfate*)	9 ± 2	9 ± 3
3-(3′-Methoxyphenyl)propanoic acid-4′-glucuronide (*DHFA- 4′-glucuronide*)	5 ± 1	7 ± 1*
3-(4′-Methoxyphenyl)propanoic acid-3′-glucuronide (*DHiFA- 3′-glucuronide*)	2.9 ± 0.5	3.1 ± 0.7
3-(3′-Methoxyphenyl)propanoic acid-4′-sulfate (*DHFA- 4′-sulfate*)	10 ± 2	9 ± 2
3-(4′-Methoxyphenyl)propanoic acid-3′-sulfate (*DHiFA- 3′-sulfate*)	3 ± 1	2.6 ± 0.8
3-(Phenyl)propanoic acid-4′-glucuronide (*DHCoA-4′-glucuronide*)	2.1 ± 0.3	1.9 ± 0.3
3-(Phenyl)propanoic acid-4′-sulfate (*DHCoA-4′-sulfate*)	21 ± 6	21 ± 4
3-Dihydrocaffeoylquinic acid	0.35 ± 0.07	0.27 ± 0.08 **
5-Dihydrocaffeoylquinic acid	0.03 ± 0.02	0.03 ± 0.01
4-Dihydrocaffeoylquinic acid	0.06 ± 0.02	0.04 ± 0.02
3-Dihydroferuloylquinic acid	0.3 ± 0.1	0.5 ± 0.07
5-Dihydroferuloylquinic acid	0.3 ± 0.1	0.18 ± 0.07
4-Dihydroferuloylquinic acid	0.06 ± 0.02	0.05 ± 0.02
Dihydrocoumaroylquinic acid	0.5 ± 0.1	0.4 ± 0.1
Dihydrocoumaroylquinic acid	0.22 ± 0.08	0.17 ± 0.07 *
Feruloylglycine	19 ± 5	25 ± 5
*Iso*Feruloylglicine	0.44 ± 0.04	0.6 ± 0.1
**TOTAL—Colonic metabolites**	106 ± 16	132 ± 17
**Other microbial metabolites**		
3′,4′-Dihydroxyphenylacetic acid	1.2 ± 0.2	1.3 ± 0.2 *
4′-Hydroxy-3′-methoxyphenylacetic acid	13.2 ± 0.9	12 ± 1
3′-Hydroxyphenylacetic acid	12 ± 3	9 ± 2
3,4-Dihydroxybenzoic acid	0.13 ± 0.02	0.18 ± 0.04
4-Hydroxybenzoic acid	1.3 ± 0.2	1.01 ± 0.09 *
3-Hydroxybenzoic acid	0.91 ± 0.07	1.0 ± 0.1
4′-Hydroxyhippuric acid	14 ± 2	14 ± 2
3′-Hydroxyhippuric acid	32 ± 5	46 ± 8 **
**TOTAL—Other microbial metabolites**	75 ± 7	85 ± 8
**TOTAL Colonic + other microbial met.**	181 ± 21	217 ± 21
**TOTAL INTESTINAL + COLONIC + OTHERS**	231 ± 26	274 ± 30

Values are means ± SEM (*n* = 9). * *p* < 0.05 ** *p* < 0.01 week 0 vs. week 8.

**Table 3 nutrients-14-02445-t003:** Amount of fecal metabolites excreted at 0 h and 24 h after consumption of the GCPE nutraceutical at the baseline (week 0) and the end of the intervention (week 8).

	0 h (µmol/g)		24 h (µmol/g)	
Metabolite	Week 0	Week 8	Week 0	Week 8
**Intestinal absorption**				
5-Feruloylquinic acid	0.0009 ± 0.0007	0.002 ± 0.002	0.002 ± 0.001	0.0003 ± 0.0003
3-Feruloylquinic acid	0.003 ± 0.003	0.0010 ± 0.0007	0.003 ± 0.003	0.003 ± 0.002
4-Feruloylquinic acid	N.D.	0.001 ± 0.001	N.D.	0.0004 ± 0.0004
3′,4′-Dihydroxycinnamic acid (*Caffeic acid*, *CA*)	N.D.	0.003 ± 0.002	0.0003 ± 0.0003	0.003 ± 0.003
4′-Hydroxy-3′-methoxycinnamic acid (*Ferulic acid*, *FA*)	0.005 ± 0.002	0.009 ± 0.003	0.011 ± 0.007	0.019 ± 0.007
4′-Hydroxycinnamic acid *(Coumaric acid*, *CoA)*	N.D.	0.006 ± 0.004	0.002 ± 0.002	0.003 ± 0.001
**TOTAL—Intestinal metabolites**	0.009 ± 0.004	0.022 ± 0.007	0.017 ± 0.008	0.028 ± 0.009
**Colonic absorption**				
3-(3′,4′-Dihydroxyphenyl)propanoic acid (*Dihydrocaffeic acid*, *DHCA*)	0.011 ± 0.007	0.005 ± 0.002	0.008 ± 0.006	0.0001 ± 0.0001
3-(4′-Hydroxy-3′-methoxyphenyl)propanoic acid (*Dihydroferulic*, *DHFA*)	0.013 ± 0.004	0.018 ± 0.003	0.019 ± 0.007	0.014 ± 0.003
3-(4′-Hydroxyphenyl)propanoic acid (*Dihydrocoumaric acid*, *DHCoA*)	0.005 ± 0.003	0.006 ± 0.006	0.005 ± 0.005	0.014 ± 0.005
3-(3′-Hydroxyphenyl)propanoic acid (*Dihydroisocoumaric acid*, *DHiCoA*)	0.21 ± 0.07	0.4 ± 0.1	0.24 ± 0.07	0.25 ± 0.07
5-Dihydrocaffeoylquinic acid	0.04 ± 0.01	0.06 ± 0.02	0.05 ± 0.01	0.08 ± 0.02
**TOTAL—Colonic metabolites**	0.28 ± 0.07	0.5 ± 0.1	0.33 ± 0.08	0.36 ± 0.08
**Other microbial metabolites**				
3′,4′-Dihydroxyphenylacetic acid	N.D.	N.D.	N.D.	0.010 ± 0.008
4′-Hydroxy-3′-methoxyphenylacetic acid	0.007 ± 0.005	0.0006 ± 0.0006	0.02 ± 0.01	0.003 ± 0.003
3′-Hydroxyphenylacetic acid	0.31 ± 0.09	0.18 ± 0.08	0.21 ± 0.08	0.11 ± 0.04
3,4-Dihydroxybenzoic acid	0.016 ± 0.006 *	0.026 ± 0.008 *	0.018 ± 0.005	0.04 ± 0.01
4-Hydroxy-3-methoxybenzoic acid	0.06 ± 0.03	0.006 ± 0.006	0.09 ± 0.05	N.D.
4-Hydroxybenzoic acid	0.004 ± 0.004	0.006 ± 0.004	0.0006 ± 0.0004	0.01 ± 0.01
3-Hydroxybenzoic acid	0.03± 0.02	0.020 ± 0.008	0.03 ± 0.01	0.020 ± 0.007
**TOTAL—Other microbial metabolites**	0.4 ± 0.1	0.24 ± 0.09	0.4 ± 0.1	0.19 ± 0.07
**TOTAL Colonic + other microbial met.**	0.7 ± 0.2	0.8 ± 0.2	0.7 ± 0.1	0.6 ± 0.1
**TOTAL INTESTINAL + COLONIC + OTHERS**	0.7 ± 0.2	0.8 ± 0.2	0.7 ± 0.1	0.6 ± 0.1

Values are means ± SEM (*n* = 9). * *p* < 0.05 week 0 vs. week 8. N.D.: Not detected.

## Data Availability

Data have not been deposited in any open repository, but they are available upon request to the authors.

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
