# Peer review of "Sustained Consumption of a Decaffeinated Green Coffee Nutraceutical Has Limited Effects on Phenolic Metabolism and Bioavailability in Overweight/Obese Subjects"

_nutrients, 2022, doi:10.3390/nu14122445_

Round 1
Reviewer 1 Report
The aim of this manuscript is to demonstrate the effect of green coffee extract on the phenol metabolism in obese populations. Nine overweight/obese volunteers were involved in this eight weeks study and result suggested that phenolic compounds mainly metabolized through the GI tract.
1. General information of study populations should be provided, including at least Age, sex, and BMI
2. Population with normal BMI should included in this study as control group
3. Is there any effect of GCPE on this populations after 2 months consumption?
4. Is there any clinic indication from this study?
5. Vital sign and important laboratory result in the study group should provided before and after 2 months
Author Response
Response to Reviewer 1 Comments
The authors would like to thank the editor and referees for their revision. We feel that their comments have greatly contributed to improving our work. The changes made in the manuscript have been shaded in yellow (referee 1) and green (referee 2). Some parts have been crossed out but not removed, so that the reduction we have made could be easily identified.
Point 1: General information of study populations should be provided, including at least Age, sex, and BMI
Response 1: Thank you for your indication. We have added the general information of the study population in section: 2.3. Participants, study design and sample collection (shaded in yellow).
Point 2: Population with normal BMI should included in this study as control group
Response 2: We agree with the referee that it would be interesting to assess the potential effects of sustained consumption of green coffee phenolic compound (GCPE) in healthy volunteers, since the literature only reports the bioavailability and metabolism of phenolic compounds after a single dose. However, as mentioned in section 2.3. of the manuscript, this study was part of a larger intervention carried out in 29 subjects with overweight or obesity, an intervention designed to assess the effect of sustained consumption of GCPE on weight control, blood pressure, lipid metabolism and glucose homeostasis. Of these participants, nine volunteers agreed to be included in the present bioavailability study, which aimed at investigating the effects of regularly consuming a green coffee phenolic extract (GCPE) on the bioavailability and metabolism of (poly)phenols. Therefore, the present study did not attempt to compare the health effects of sustained consumption of GPCE between obese and healthy populations. We are aware that the microbiota composition and, therefore, its potential metabolic capacity might be altered in the overweight/obese volunteers, thus the interest of performing such long-term study in healthy subjects, which are garanteed in the future, but were not the aim of the present manuscript.
Point 3: Is there any effect of GCPE on this populations after 2 months consumption?
Response 3: Thank you for the observation. Regarding health effects, no adverse events were reported. The main biochemical data and vital signs of the participants at baseline and week 8 have been included as suggested in your coment 5. Based on these data, there are no statistical differences between weeks 0 and 8 of the intervention in these nine participants. However, a slight tendency to improve lipid profile, body weight and systolic blood pressure was observed after 8 weeks of GCPE consumption.
The effect of sustained consumption on bioavailability is detailed in the discussion section of the manuscript, and it is concluded that a higher trend in the absorption of GCPE was observed after regular consumption, but there were no statistically significant changes in the metabolic profiles in plasma, urine, and feces.
Point 4: Is there any clinic indication from this study?
Response 4: Thank you. This aspect has been answered in the previous point.
Point 5: Vital sign and important laboratory result in the study group should provided before and after 2 months
Response 5: Thank you for your comment. The main biochemical and vital characteristics of the volunteers at baseline (week 0) and at the end of the intervention (week 8) are shown in Supplementary Table S2. (This indication is given in the manuscript shaded in yellow).
Reviewer 2 Report
The authors investigated the effects of regularly consuming green coffee polyphenolic extract (GCPE) on the bioavailability and metabolisms. They compared the metabolites from blood, urine, and feces samples obtained from nine participants at baseline and after eight weeks of consumption of GCPE. Although some metabolites of polyphenolic compounds in the urine and feces samples were changed and a higher trend in the absorption was observed after eight weeks of GCPE consumption, the overall effect on the bioavailability of the polyphenol metabolites was small.
The reviewer thinks that the experiments and primary structure of the manuscript are well organized. However, some minor concerns that the authors should respond to exist.
1. Line 59, ref 18 was published in 2019 in the reference section; however, it was published in 2021 in the main text. Which is correct?
2. Line 196-199, they confirmed data normality by Shapiro-Wilk test. Why did they take the non-parametric test instead of the parametric paired t-test?
3. Table 1 would be better to juxtapose each parameter data after eight weeks of GCPE consumption besides the baseline data.
4. Figure 3 would be better to use a boxplot with overlaid dot plot as the number of participants was only 9.
5. Lines 564-578, they discussed inter-individual differences of metabolites. Add some example data as the supplement information and discuss more specifically. Data not shown in line 565 is not preferable.
6. Is it possible to add the physiological meaning of the long-term GCPE consumption in the present study in the conclusion section? At present, the reviewer wonders, so what?
Author Response
The authors would like to thank the editor and referees for their revision. We feel that their comments have greatly contributed to improving our work. The changes made in the manuscript have been shaded in yellow (referee 1) and green (referee 2). Some parts have been crossed out but not removed, so that the reduction we have made could be easily identified.
Point 1: Line 59, ref 18 was published in 2019 in the reference section; however, it was published in 2021 in the main text. Which is correct?
Response 1: We apologize for this mistake. References in the main text are correct (reference 18 was published in 2021 while reference 19 was published in 2019). A swap error occurred in the compilation of the reference list. In agreement with your comment, the reference section has been corrected (shaded in green).
Point 2: Line 196-199, they confirmed data normality by Shapiro-Wilk test. Why did they take the non-parametric test instead of the parametric paired t-test?
Response 2: Thank you for your comment. There was a mistake in the interpretation of the sentence. We meant to say that data normality was tested using the Shapiro Wilk test, and in the absence of normality, we performed non-parametric Wilcoxon test for paired comparisons. The Shapiro–Wilk test is a more appropriate method to assess data normality in small sample sizes (<50 samples) while Kolmogorov-Smirnov test is used for n ≥ 50. *
We have rewritten the next sentences in the manuscript (shaded in green) to avoid misinterpretation: “The Shapiro-Wilk test was used to assess data normality. In view of the lack of normality and considering the small sample size, comparisons between week-0 and week-8 were performed by the non-parametric Wilcoxon test for paired comparisons.”
*Mishra, P., Pandey, C. M., Singh, U., Gupta, A., Sahu, C., & Keshri, A. (2019). Descriptive statistics and normality tests for statistical data. Annals of cardiac anaesthesia, 22(1), 67–72. https://doi.org/10.4103/aca.ACA_157_18
Point 3: Table 1 would be better to juxtapose each parameter data after eight weeks of GCPE consumption besides the baseline data.
Response 3: Thank you. The Cmax, Tmax and AUC0-24h parameters at baseline and after 8-weeks of GPCE consumption have been reorganized in Table 1. Tables 2 and 3 have also been reorganized according to the proposed new format.
Point 4: Figure 3 would be better to use a boxplot with overlaid dot plot as the number of participants was only 9.
Response 4: Thank you for the observation. The bar chart in figure 3B has been replaced by a dot plot graph.
Point 5: Lines 564-578, they discussed inter-individual differences of metabolites. Add some example data as the supplement information and discuss more specifically. Data not shown in line 565 is not preferable.
Response 5: Regarding inter-individual differences, we have included in the Supplementary Information a detailed study of the inter-individual differences in plasma, urine and faeces associated with GCPE nutraceutical consumption. We have also added a brief explanation in the manuscript on this issue (shaded in green).
Point 6: Is it possible to add the physiological meaning of the long-term GCPE consumption in the present study in the conclusion section? At present, the reviewer wonders, so what?
Response 6: Thank you for your indication. Specific physiological effects with implications for health have been attributed to phenol-rich extracts as GCPE (anti-hypertensive, anti-hyperlipidemic, anti-fibrotic or anti-inflammatory activities). Despite intense and active research investigating the health effects of plant food bioactives, the real impact of most compounds and their mechanisms of action depends of their bioavailability and metabolic fate. This study aimed at investigating the effects of regularly consuming a green coffee phenolic extract (GCPE) on the bioavailability and metabolism of (poly)phenols, so the clinical importance of these findings is being addressed for future work. The next paragraph has been added in the conclusion section (shaded in green) to elucidate this issue: “The present study contributes to better understand the effect of sustained consumption of a phenol-rich extract such as GCPE, since there are limited data on repeated exposure to phenols in bioavailability studies. This could help to develop refined dietary strategies and recommendations to optimize the beneficial effects of phenol-rich foods”
Round 2
Reviewer 1 Report
The authors appropriate response the reviewer's comments, with much improvement of the manuscript.
Author Response
Thank you very much for your guidance